# Antidepressant use and all-cause mortality in depressed individuals: A real-world cohort study

**Shaoyu Zhou** [1]*, **Caixia Wang**[2], **Yanping Zhang**[3]*

**1** Department of Psychiatry, Jinshan Mental Health Center, Shanghai, China, **2** Medical Department, Jinshan Mental Health Center, Shanghai, China, **3** Hospital Office of Mental Health Sub-center of Jinshan Disease Prevention and Control Center, Shanghai, China

* zhoushaoyu7926@163.com (SZ); 15221020109@fudan.edu.cn (YZ)

## Abstract

### Background

While antidepressants are effective in alleviating symptoms, their association with mortality remains unclear. This research investigated the link between antidepressant usage and all-cause mortality among depressed patients.

### Methods

We performed a real-world study on 5,947 adults with depression using a dataset from the National Health and Nutrition Examination Survey (2005–2018). Depression was identified by a Patient Health Questionnaire-9 score ≥10, or the use of antidepressants, with all-cause mortality assessed through the National Death Index. Covariates included demographics, socioeconomic status, lifestyle factors, and chronic conditions. The study performed weighted Cox proportional-hazards models, propensity score methods, and inverse probability of treatment weighting (IPTW) to estimate hazard ratios (HRs) and 95% confidence intervals (CIs) for comparing mortality risk between patients treated with antidepressants and those who were not. We conducted sensitivity analyses to evaluate the robustness of our findings.

### Results

During the median 82-month follow-up period, 15.0% of participants (n = 894) died. Antidepressant users (n = 3,925) had a crude mortality rate of 16.5%, compared to 12.2% in non-users (n = 2,022). The crude Cox proportional-hazards analysis indicated that antidepressant use was linked to a non-significant elevation in mortality (HR = 1.18, 95% CI 0.95–1.47, P = 0.126). This association attenuated completely after covariate adjustment (adjusted HR = 0.92, 95% CI 0.75–1.13). Propensity score analyses indicated no significant link between antidepressant use and mortality (IPTW, HR = 0.96, 95% CI 0.80–1.16, P = 0.707). Across all methods, no statistically significant association was observed.

**Data availability statement:** The original data used in this study are publicly available from the National Health and Nutrition Examination Survey (NHANES) database (https://www.cdc.gov/nchs/nhanes/). The processed datasets supporting the findings of this study have been included as Supporting Information files with this manuscript.

**Funding:** The author(s) received no specific funding for this work.

**Competing interests:** The authors have declared that no competing interests exist.

## Conclusion

All-cause mortality is not significantly affected by the overall use of antidepressants in individuals with depression; however, future studies should investigate safety differences between specific drug classes.

## Introduction

An estimated 280 million individuals globally, representing approximately 5% of the adult population, are affected by depression [1]. Depression is a major contributor to global disability and disease burden. In 2019, depressive disorders were responsible for around 46.9 million Disability-Adjusted Life Years (DALYs), marking a significant rise since 1990 [2,3]. Individuals with depression face increased risks for various physical health conditions. They also have a higher likelihood of experiencing complications from these conditions, which can lead to premature mortality [2]. Depression is a significant contributor to suicide risk and ranks among the leading causes of death for young people worldwide [4].

Antidepressants have been empirically validated to alleviate depressive symptoms, as supported by a meta-analysis involving more than 100,000 patients. This analysis showcased significant therapeutic effectiveness across different types of antidepressants, wherein odds ratios consistently favored active treatment over placebos [5]. Empirical studies consistently demonstrated a strong link between depression and elevated risk of all-cause mortality [6,7]. Antidepressants may not only alleviate depressive symptoms but also potentially lower all-cause mortality. However, some studies have suggested a potential association between antidepressant use and increased mortality risk [8,9], raising questions about their overall benefit. Various factors, such as disease severity, comorbidities, and lifestyle choices, may have a greater impact on mortality outcomes. Thus, attributing mortality risk solely to antidepressant use may oversimplify the issue. Some studies indicate that antidepressants may lower mortality in certain groups, such as patients with hepatocellular carcinoma who use antidepressants post-diagnosis [10]. Nevertheless, the effect of antidepressants on mortality in the depressed population remains unclear.

We performed a real-world cohort study to explore the link between antidepressant usage and all-cause mortality in adults with depression.

## Materials and methods

### Study participants

The real-world study was conducted from 2005 to 2018, with a follow-up ending in March 2019. The study population was sourced from the NHANES database, focusing on individuals diagnosed with depression. This study examined a nationally representative sample of 5,947 adults aged 20 and older, sourced from seven consecutive NHANES cycles. To ensure data integrity and analytical rigor, the study implemented stringent exclusion criteria, eliminating participants with incomplete records in key variables including psychological health evaluations (depression screening metrics),

sociodemographic characteristics (marital status and educational attainment), behavioral factors (tobacco use patterns), anthropometric measurements (body mass index), and mortality outcomes. The exclusion protocol was designed to maintain methodological consistency while minimizing potential confounding factors in subsequent analyses. All data were collected through standardized NHANES protocols, including questionnaires, physical examinations, and laboratory tests.

## Depression assessment methodology

The operational definition of depression in this study incorporated both psychometric evaluation and pharmacological treatment indicators. Case ascertainment was determined by a Patient Health Questionnaire-9 (PHQ-9) score of ≥10 or recorded use of antidepressants [11]. The PHQ-9, a validated tool for screening depressive disorders, assesses symptom frequency based on nine DSM-5 criteria over two weeks, using a 4-point Likert scale (0 = not at all to 3 = nearly every day). This scoring system yields a composite range of 0–27, with the established clinical cutoff of ≥10 demonstrating 88% sensitivity and specificity for major depressive disorder in validation studies [12]. Concurrent with psychometric assessment, antidepressant usage was determined through a comprehensive review of prescription medication records within the NHANES database, complemented by participant self-reports during baseline interviews. This dual-criterion approach (combining symptomatic presentation and treatment status) was adopted as the primary case definition to enhance diagnostic accuracy and ensure a comprehensive capture of depression cases across the clinical spectrum, from newly identified to currently managed conditions.

Given that certain prior studies solely utilized PHQ-9 scores of ≥10 for depression diagnosis [13,14], we incorporated this criterion as an additional measure in our sensitivity analyses to enhance the robustness of our research outcomes.

## Covariates

The covariates were selected based on their potential associations with depressive symptoms, antidepressant use, and all-cause mortality [11,15]. The covariates comprised demographic factors including age (as a continuous variable), gender, racial/ethnic groups (non-Hispanic White, non-Hispanic Black, others), educational level (<high school, high school diploma/equivalent, ≥ college degree), marital/cohabitation status (never married, married/cohabiting, others), and socioeconomic status assessed by the poverty-income ratio (PIR ≤ 1.3, 1.3–3.5, > 3.5, unknown). Additionally, lifestyle factors such as alcohol consumption, smoking status (never, former, and current), and physical activity were considered. Body mass index (BMI) categories were defined as normal/underweight, overweight, and obese. Chronic diseases identified include diabetes, cardiovascular disease, hypertension, arthritis, chronic kidney disease, and cancer.CVD includes conditions like heart failure, myocardial infarction, coronary artery disease, stroke, and congestive heart failure [15]. Diabetes mellitus is defined by a glycated hemoglobin level of 6.5% or higher, a fasting plasma glucose level exceeding 7.0 mmol/L, or self-reported diabetes requiring insulin [16]. Chronic kidney disease (CKD) was diagnosed when the urinary albumin-to-creatinine ratio was over 30 mg/g or the estimated glomerular filtration rate (eGFR) was under 60 mL/min/1.73 m² [17]. A history of arthritis and cancer was self-reported [18,19].

## All-cause mortality

All-cause mortality status was determined by probabilistically matching NHANES participant records with the National Death Index using identifying variables such as name, social security number, and date of birth. The mortality follow-up period extended through December 31, 2019, with death outcomes being validated through official death certificate records. The National Center for Health Statistics (NCHS) executed this linkage methodology, known for its high sensitivity and specificity in prior validation studies, in accordance with established protocols. Additional methodological details regarding the mortality data linkage process, including matching algorithms and quality control measures, are available through the NCHS data linkage resource portal (accessible at www.cdc.gov/nchs/data-linkage/mortality.htm).

### Ethics approval statement

This study employed anonymized participant data from the publicly accessible NHANES database. The NHANES protocol was approved by the National Center for Health Statistics (NCHS) Research Ethics Review Board (ERB), ensuring that the study design and implementation met ethical requirements.

### Statistical analysis

Participant characteristics were summarized using descriptive statistics appropriate to the data type. Continuous variables were represented as means with standard deviations, and categorical variables were shown as frequencies and percentages. Group differences were evaluated using Kruskal-Wallis tests for continuous variables and Chi-square tests for categorical variables. In the initial phase of analysis, hazard ratios (HRs) and corresponding confidence intervals (CIs) were derived using a weighted Cox proportional hazards model without covariate adjustment to establish crude estimates. Subsequently, to account for potential confounders, we computed adjusted HRs and CIs using a weighted multivariate Cox proportional hazards model incorporating all relevant covariates. To further account for the imbalance in group sizes and confounding, we employed propensity score methods. A logistic regression model incorporating all covariates was used to estimate the propensity score. We employed three methods to estimate treatment effects, including incorporating the propensity score as a covariate in a multivariable Cox proportional-hazards model. Inverse Probability of Treatment Weighting (IPTW) was employed as the primary analysis method. Utilizing the propensity score, weights were calculated to estimate the average treatment effect for the treated (ATT), the control (ATC), and the entire population (ATE). Patients were matched using propensity scores, and the matched dataset was analyzed with a weighted multivariate Cox proportional hazards model to estimate ATT, ATC, and ATE [20–22].

We subsequently reanalyzed the preceding investigation utilizing the sensitivity analysis definition of depression (PHQ-9 ≥ 10) to validate the robustness of the results.

Statistical significance was assessed using two-sided P values. Analyses were performed using EmpowerStats (www.empowerstats.com) and R software version 4.2.0.

## Results

### Participant selection and baseline characteristics

From a total of 70,190 patients screened over seven cycles, 6,280 individuals aged 20 and older exhibited depression. After excluding 333 patients due to missing data on marital status, education level, smoking status, and mortality, a final cohort of 5,947 individuals with depression was included in the study (Fig 1). Among these, 3,925 patients (66.0%) received antidepressant treatment, while 2,022 patients (34.0%) did not.

Table 1 displays the baseline characteristics of patients based on antidepressant use, comparing both unmatched samples and those analyzed with propensity score matching (PSM). Before PSM, significant differences were observed between the antidepressant-treated group (n = 3,925) and the untreated group (n = 2,022) in demographics, socioeconomic status, lifestyle, and chronic disease prevalence (all P < 0.05). The antidepressant group was characterized by an older average age (55.3 ± 16.0 years compared to 47.4 ± 16.8 years, P < 0.001), a greater percentage of females (67.6% versus 60.4%, P < 0.001), a higher representation of non-Hispanic Whites (64.6% against 35.9%, P < 0.001), and a more significant prevalence of chronic diseases, such as hypertension (53.1% compared to 42.7%, P < 0.001). After PSM (n = 891 per group), covariates including age (52.04 ± 14.29 vs. 51.82 ± 16.58 years, P = 0.767), gender (male 30.42% vs. 31.09%, P = 0.758), education level (college or above 45.79% vs. 44.78%, P = 0.574), and chronic conditions (e.g., diabetes 28.40% vs. 29.63%, P = 0.566) were well-balanced. However, residual differences in race distribution persisted (non-Hispanic whites 53.54% vs. 48.15%, P < 0.001). Before matching, HDL-C levels showed significant differences, but these differences were no longer statistically significant post-matching. Furthermore, no notable variations were observed

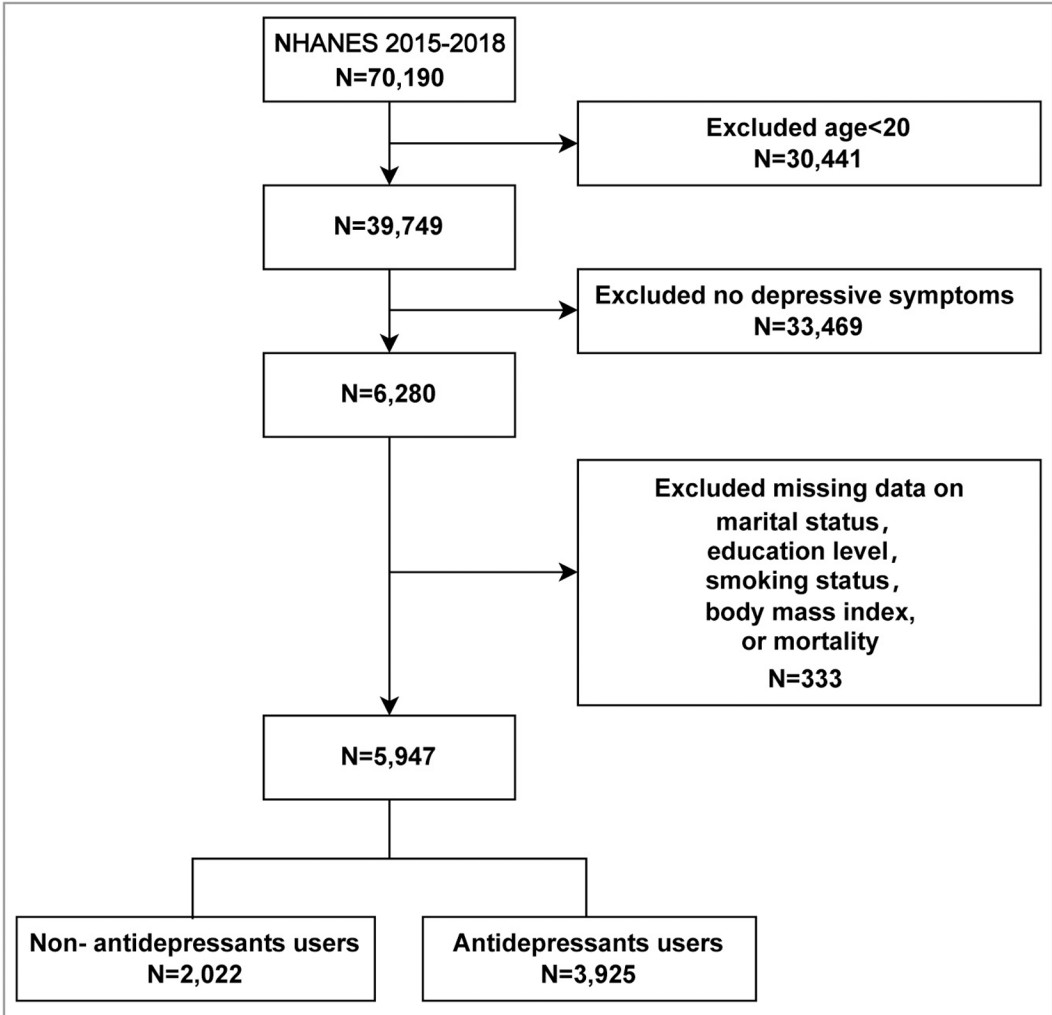

**Fig 1. Flow chart of sample selection.**

in other lipid parameters, such as total cholesterol, triglycerides, and LDL-C, either before or after matching (all P > 0.05). PSM effectively balanced the baseline characteristics between the antidepressant-treated and untreated groups, indicating that the matching method successfully controlled for potential confounding factors.

## Association between antidepressant use and mortality risk

Among the 5,947 individuals with depression, 894 deaths (15.0%) occurred during the median follow-up period of 82 months. Although the crude mortality rate appeared higher in antidepressant users (647/3,925, 16.5%) compared to non-users (247/2,022, 12.2%), this imbalance reflects the real-world distribution of antidepressant use among individuals with depression in the NHANES dataset. The higher crude mortality rate in antidepressant users can be largely attributed to significant baseline differences between the groups, including older age (55.3 ± 16.0 vs. 47.4 ± 16.8 years, P < 0.001) and higher prevalence of chronic conditions in the antidepressant group. To address this potential confounding, we employed multiple statistical approaches including multivariable adjustment, propensity score matching, and IPTW. Table 2 indicates that, although antidepressant users exhibited an elevated risk of mortality compared to non-users in the

**Table 1. Characteristics of patients receiving or not receiving antidepressants, before and after propensity-score matching.**

| | Unmatched Patients | | | Propensity-Score–Matched Patients | | |
|---|---|---|---|---|---|---|
| | No Antidepressants | Antidepressants | P value | No Antidepressants | Antidepressants | P value |
| N | 2022 | 3925 | | 891 | 891 | |
| Age, years | 47.4±16.8 | 55.3±16.0 | <0.001 | 51.82±16.58 | 52.04±14.29 | 0.767 |
| Gender, n (Male, %) | 801 (39.6%) | 1271 (32.4%) | <0.001 | 277 (31.09%) | 271 (30.42%) | 0.758 |
| Racial/ethnic, n (%) | | | <0.001 | | | <0.001 |
| Non-Hispanic White | 726 (35.9%) | 2537 (64.6%) | | 429 (48.15%) | 477 (53.54%) | |
| Non-Hispanic Black | 502 (24.8%) | 543 (13.8%) | | 227 (25.48%) | 156 (17.51%) | |
| Others | 794 (39.3%) | 845 (21.5%) | | 235 (26.37%) | 258 (28.96%) | |
| Education, n (%) | | | <0.001 | | | 0.574 |
| Less than high school | 792 (39.2%) | 866 (22.1%) | | 286 (32.10%) | 266 (29.85%) | |
| High school or equivalent | 480 (23.7%) | 928 (23.6%) | | 206 (23.12%) | 217 (24.35%) | |
| College or above | 750 (37.1%) | 2131 (54.3%) | | 399 (44.78%) | 408 (45.79%) | |
| Marital status, n (%) | | | <0.001 | | | 0.206 |
| never_married | 459 (22.7%) | 516 (13.1%) | | 163 (18.29%) | 149 (16.72%) | |
| Married/cohabiting | 928 (45.9%) | 2115 (53.9%) | | 409 (45.90%) | 387 (43.43%) | |
| Others | 635 (31.4%) | 1294 (33.0%) | | 319 (35.80%) | 355 (39.84%) | |
| PIR, n (%) | | | <0.001 | | | 0.529 |
| ≤1.3 | 998 (49.4%) | 1250 (31.8%) | | 386 (43.32%) | 414 (46.46%) | |
| 1.3–3.5 | 607 (30.0%) | 1326 (33.8%) | | 294 (33.00%) | 271 (30.42%) | |
| >3.5 | 226 (11.2%) | 1061 (27.0%) | | 136 (15.26%) | 128 (14.37%) | |
| Unknown | 191 (9.4%) | 288 (7.3%) | | 75 (8.42%) | 78 (8.75%) | |
| Drinking, n (%) | | | <0.001 | | | 0.272 |
| No | 499 (24.7%) | 849 (21.6%) | | 247 (27.72%) | 218 (24.47%) | |
| Yes | 1253 (62.0%) | 2211 (56.3%) | | 510 (57.24%) | 527 (59.15%) | |
| Unkown | 270 (13.4%) | 865 (22.0%) | | 134 (15.04%) | 146 (16.39%) | |
| Smoking, n (%) | | | <0.001 | | | 0.472 |
| Never | 850 (42.0%) | 1735 (44.2%) | | 344 (38.61%) | 324 (36.36%) | |
| Former | 415 (20.5%) | 1182 (30.1%) | | 212 (23.79%) | 232 (26.04%) | |
| Now | 757 (37.4%) | 1008 (25.7%) | | 335 (37.60%) | 335 (37.60%) | |
| Physical activity, n (%) | 572 (28.3%) | 1279 (32.6%) | <0.001 | 224 (25.14%) | 214 (24.02%) | 0.582 |
| BMI, n (%) | | | <0.001 | | | 0.191 |
| Normal/Underweight | 548 (27.1%) | 871 (22.2%) | | 171 (19.19%) | 167 (18.74%) | |
| Overweight | 568 (28.1%) | 1120 (28.5%) | | 226 (25.36%) | 196 (22.00%) | |
| Obesity | 906 (44.8%) | 1934 (49.3%) | | 494 (55.44%) | 528 (59.26%) | |
| Type 2 diabetes, n (%) | 450 (22.3%) | 967 (24.6%) | 0.041 | 264 (29.63%) | 253 (28.40%) | 0.566 |
| CVD, n (%) | 339 (16.8%) | 822 (20.9%) | <0.001 | 216 (24.24%) | 221 (24.80%) | 0.783 |
| Hypertensive, n (%) | 863 (42.7%) | 2085 (53.1%) | <0.001 | 493 (55.33%) | 510 (57.24%) | 0.417 |
| Arthritis, n (%) | 768 (38.0%) | 2014 (51.3%) | <0.001 | 518 (58.14%) | 514 (57.69%) | 0.848 |
| CKD, n (%) | 358 (17.7%) | 848 (21.6%) | <0.001 | 168 (18.86%) | 168 (18.86%) | 1.000 |
| Cancer, n (%) | 201 (9.9%) | 607 (15.5%) | <0.001 | 116 (13.02%) | 126 (14.14%) | 0.489 |
| Death, n (%) | 247 (12.2%) | 647 (16.5%) | <0.001 | 142 (15.94%) | 123 (13.80%) | 0.206 |
| Total cholesterol, mg/dL | 194.6±44.5 | 195.7±43.6 | 0.357 | 196.12±46.11 | 196.74±44.75 | 0.779 |
| Triglyceride, mg/dL | 143.1±144.6 | 144.5±114.7 | 0.789 | 143.99±98.05 | 146.46±112.18 | 0.740 |
| LDL-cholesterol, mg/dL | 115.1±38.0 | 112.7±37.6 | 0.125 | 115.73±41.72 | 112.93±37.25 | 0.319 |
| HDL-Cholesterol, mg/dL | 51.3±16.1 | 53.6±16.9 | <0.001 | 50.90±15.22 | 51.74±15.88 | 0.268 |

N represents the number of patients in each group. PIR: Poverty Income Ratio. CVD: Cardiovascular Disease. CKD: Chronic Kidney Disease. LDL: Low-Density Lipoprotein. HDL: High-Density Lipoprotein.

crude analysis, the association lacked statistical significance (HR = 1.18, 95% CI 0.95–1.47, P = 0.126). After accounting for all covariates, the HR was 0.92 (95% CI 0.75–1.13, P = 0.435), suggesting a possible reduction in mortality risk for antidepressant users, but the result was not statistically significant. Incorporating the propensity score into the multi-variable Cox proportional-hazards model yielded an HR of 1.03 (95% CI 0.87–1.21, P = 0.726), indicating no significant link between antidepressant use and mortality. Using IPTW to estimate treatment effects, the HR for ATT, ATC, and ATE were 0.96 (95% CI: 0.80–1.16, P = 0.707), 1.13 (95% CI: 0.94–1.35, P = 0.187), and 1.02 (95% CI: 0.85–1.21, P = 0.863), respectively. The findings suggest that mortality risk does not significantly differ between antidepressant users and non-users across various populations. In the PSM analysis, the HR for ATT, ATC, and ATE were 0.83 (95% CI: 0.65–1.06, P = 0.137), 1.17 (95% CI: 0.94–1.46, P = 0.151), and 0.93 (95% CI: 0.75–1.15, P = 0.479), respectively. Although antide-pressant use showed a trend towards a protective effect in the treated group (ATT HR < 1) and a marginally increased risk in the control group (ATC HR > 1), these associations were not statistically significant.

The sensitivity analysis, using PHQ-9 ≥ 10, confirmed the primary analysis results, demonstrating robustness: IPTW HR = 0.98 (95% CI 0.78–1.24, P = 0.889), ATT HR = 1.12 (95% CI 0.85–1.46, P = 0.419), and ATE HR = 1.07 (95% CI 0.84–1.36, P = 0.579). Supplementary material S1 and S2 Tables in S1 File provide detailed sensitivity analysis results.

Overall, there was no statistically significant association detected between the use of antidepressants and the risk of mortality across all analytical methods.

## Discussion

Our study found no statistically significant association between the overall use of antidepressants and all-cause mortal-ity after comprehensive adjustment for potential confounding factors. While the unadjusted analysis showed a slightly higher crude mortality rate among antidepressant users (16.5% vs. 12.2%), this difference was not statistically significant (HR = 1.18, 95% CI 0.95–1.47, P = 0.126). After adjusting for relevant covariates, the hazard ratio decreased to 0.92 (95% CI 0.75–1.13, P = 0.435), suggesting that the initial trend may have been influenced by differences in baseline charac-teristics between the groups. Our propensity score analyses further confirmed the absence of a significant association between overall antidepressant use and all-cause mortality risk. Sensitivity analyses, which defined depression strictly based on depressive symptoms (PHQ-9 ≥ 10), produced similar findings. These findings underscore an important meth-odological consideration in real-world research: the apparent differences in crude mortality rates between groups may be largely explained by confounding factors. Our results highlight the necessity of appropriate statistical adjustment, especially when baseline characteristics differ substantially between comparison groups. Collectively, our study provides important evidence regarding the overall safety of antidepressants with respect to mortality outcomes in individuals with depression.

Our findings align partially with existing literature, yet discrepancies highlight the complex relationship between anti-depressant use and mortality risk. For instance, a 2017 meta-analysis reported elevated mortality risk in the general population with antidepressant use (HR = 1.33, 95% CI: 1.14–1.55), yet the relationship was not observed in individuals with cardiovascular disease [23]. This suggests that the effects of antidepressants may vary depending on the character-istics of the population. Interestingly, in certain populations, antidepressants may even reduce mortality risk. For instance, Huang et al. Post-diagnosis use of antidepressants in hepatocellular carcinoma patients was linked to a decrease in mor-tality (HR = 0.69, 95% CI 0.68–0.70) [10]. Similarly, Orayj et al.found that antidepressant use might reduce mortality rates in Parkinson's disease patients [24]. Additionally, a study on African Americans reported that the underuse of antidepres-sants in this population was associated with increased mortality [25]. However, other studies have shown that antidepres-sants may increase mortality risk in certain populations. As demonstrated by Jeffery et al. Antidepressant use was linked to a notably increased mortality risk in individuals with both depression and type 2 diabetes (HR = 2.77, 95% CI 2.48–3.10) [9]. This finding differs from our null association, potentially due to variations in study populations. Jeffery et al.focused on a high-risk group with both depression and diabetes, where the severity of depression and physical comorbidities likely

**Table 2. Associations between antidepressant use and death in the crude analysis, multivariable analysis, and propensity-score analyses.**

| Analysis | Death |
|---|---|
| No. of events/no. of patients at risk (%) | |
| Antidepressant Users | 647/3925 (16.5%) |
| Non-Antidepressant Users | 247/2022 (12.2%) |
| Crude analysis — hazard ratio (95% CI) P value | 1.18(0.95-1.47)0.126 |
| Adjust for all covariates* — hazard ratio (95% CI) P value | 0.92(0.75-1.13)0.435 |
| Adjust for PS* — hazard ratio (95% CI) P value | 1.03(0.87, 1.21)0.726 |
| Estimate of treatment effect using IPTW — hazard ratio (95% CI) P value† | |
| ATT | 0.96 (0.80, 1.16) 0.707 |
| ATC | 1.13 (0.94, 1.35) 0.186 |
| ATE | 1.02 (0.85, 1.21) 0.863 |
| Estimate of treatment effect using PS match—hazard ratio (95% CI) P value† | |
| ATT | 0.83 (0.65, 1.06) 0.137 |
| ATC | 1.17 (0.94, 1.46) 0.151 |
| ATE | 0.93 (0.75, 1.15) 0.479 |

PS: propensity score, ATT: average treatment effect for treated, ATC: average treatment effect for control, ATE: average treatment effect for all.

*Shown is the hazard ratio from the weighted multivariable Cox proportional-hazards model, with adjustment for age, gender, race, education level, marital status, and the poverty-income ratio, alcohol consumption, smoking status, physical activity, Body mass index, diabetes, cardiovascular disease, hypertension, arthritis, chronic kidney disease, and cancer. The analysis included all 5947 patients. The propensity score is estimated by incorporating all the aforementioned covariates.

†Shown is the analysis with hazard ratio from the multivariable Cox proportional-hazards model in the matched data with matching the propensity score or inverse probability of treatment weighting using the propensity score.

played a more pronounced role in mortality. In contrast, our study included a broader, nationally representative cohort of individuals with depression, capturing a wider range of depression severity and comorbidities. Similarly, Ön et al.reported increased risks of stroke and mortality in elderly antidepressant users [8]. The difference could be due to the older age and increased baseline cardiovascular risk in the cohort studied by Ön et al.

It should be emphasized that our findings reflect the average effect of antidepressant use as a whole and should not be construed as evidence that all classes of antidepressants share identical safety profiles. In this study, antidepressants were analyzed collectively, without differentiation by specific class. However, various classes—such as selective serotonin reuptake inhibitors (SSRIs), serotonin-norepinephrine reuptake inhibitors (SNRIs), tricyclic antidepressants (TCAs), and monoamine oxidase inhibitors (MAOIs)—possess distinct pharmacological characteristics and may present differing safety concerns. For example, TCAs have been associated with a higher risk of cardiovascular adverse events compared to some other antidepressants [26,27]. Additionally, our study faced several methodological challenges worth noting. The assessment of depression using PHQ-9 or antidepressant use, while practical for large epidemiological studies, may not capture the nuanced clinical presentation of depression. Moreover, the NHANES database lacks detailed information on antidepressant dosage, duration, and adherence patterns, which may influence mortality outcomes. These limitations may hinder causal inference, despite the application of robust statistical methods to control for confounders.

Our findings have significant clinical implications. This study provides an important reference for clinicians in their decision-making regarding antidepressant prescription. The discovery that antidepressant use does not significantly

correlate with all-cause mortality may ease safety concerns for healthcare providers and patients. For clinical practice, while antidepressants as a collective group show no significant mortality risk, clinicians should still consider individual patient factors (e.g., comorbidities, age) and drug-specific safety profiles when prescribing. Pre-treatment safety assessments remain essential, particularly for agents with known risks (e.g., TCAs in cardiovascular disease). In healthcare policy, guidelines for depression management should recognize the overall safety of antidepressants in relation to mortality outcomes, while also emphasizing the significance of appropriate prescribing practices. Mental health literacy programs should address misconceptions about the safety of antidepressants to diminish stigma and overcome barriers to treatment. Future research directions may include exploring the long-term safety of different types of antidepressants and evaluating their benefit-risk ratios in specific populations.

This study has several strengths that enhance the reliability and clinical relevance of its findings. First, it utilized the NHANES database, a nationally representative and longitudinal health survey, ensuring a diverse and comprehensive sample. This broad representation of individuals across different ages, races, socioeconomic statuses, and health conditions increases the external validity of the results and their applicability to real-world settings. Second, the study employed a rigorous research design and analysis strategies. The study employed PSM, IPTW, and weighted multivariable Cox proportional hazards models to minimize confounding effects, enhancing causal inference. The study assessed the link between antidepressant use and mortality, while also analyzing treatment effects across various populations using ATT, ATC, and ATE methods for a more detailed perspective. Finally, the use of real-world data, combined with detailed baseline characteristics and long-term follow-up, provides valuable evidence for assessing the long-term safety of antidepressants. These strengths make this study a unique contribution to the existing literature and an important reference for future research and clinical practice.

Despite its strengths, our study has several important limitations that warrant consideration. First, our findings are limited to adults aged 20 and older and may not be applicable to adolescents or young adults under 20, a population with unique considerations regarding antidepressant use and safety. Second, although we employed multiple statistical methods to control for confounding factors, residual confounding from unmeasured variables remains possible. Specifically, we lacked detailed information on treatment adherence, changes in depression severity over time, concurrent psychotherapy, and undocumented medications that might interact with antidepressants. Third, the NHANES database, while comprehensive, provides limited information on medication dosage, duration, and patterns of use, which may influence mortality outcomes. Fourth, our operational definition of depression using PHQ-9 scores or antidepressant use, while pragmatic for large-scale epidemiological studies, may not capture the full clinical spectrum of depressive disorders with the same precision as structured clinical interviews. Fifth, the study did not differentiate between different types of antidepressants, limiting our ability to assess the safety of specific antidepressant classes. This is particularly important as different antidepressants—such as SSRIs, SNRIs, TCAs, and MAOIs—have distinct pharmacological profiles and potentially different safety considerations. Sixth, we could not account for potential selection bias in antidepressant prescription, where physicians might preferentially prescribe certain antidepressants to patients with specific health profiles. Finally, while NHANES provides a nationally representative sample of the US population, extrapolating these results to other countries or regions requires caution, as genetic backgrounds, healthcare systems, prescribing practices, and lifestyle factors may differ substantially across populations. These limitations highlight the need for further research with more detailed medication data and longer follow-up periods to understand better the long-term safety of specific antidepressant classes in various subpopulations.

## Conclusions

Analysis of this population-based cohort revealed no statistically meaningful link between antidepressant treatment, analyzed as a collective group, and mortality risk among depressed participants. While these results offer general reassurance about antidepressant safety regarding mortality outcomes, the limitations of our study, including the inability to

differentiate between antidepressant classes and potential residual confounding, should be considered when interpreting these findings. Further research is needed to address these limitations and explore the potential benefits or risks of different antidepressants for specific subpopulations.

## Supporting information

**S1 File.** S1 Table. Characteristics of Patients (based on PHQ-9 ≥ 10) Receiving or Not Receiving Antidepressants, before and after PSM. S2 Table. Associations between Antidepressant Use and Death in the Crude Analysis, Multivariable Analysis, and Propensity-Score Analyses.
(DOCX)

## Author contributions

**Conceptualization:** Shaoyu Zhou, Yanping Zhang.

**Data curation:** Shaoyu Zhou, Caixia Wang.

**Formal analysis:** Shaoyu Zhou, Caixia Wang.

**Methodology:** Shaoyu Zhou, Caixia Wang, Yanping Zhang.

**Resources:** Yanping Zhang.

**Software:** Shaoyu Zhou.

**Supervision:** Yanping Zhang.

**Writing – original draft:** Shaoyu Zhou.

**Writing – review & editing:** Caixia Wang, Yanping Zhang.

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
