## [Decision Letter · Decision Letter 0]

PONE-D-25-19212Antidepressant Use and All-Cause Mortality in Depressed Individuals: A Real-World Cohort StudyPLOS ONE

Dear Dr. zhou,

Thank you for submitting your manuscript to PLOS ONE. After careful consideration, we feel that it has merit but does not fully meet PLOS ONE’s publication criteria as it currently stands. Therefore, we invite you to submit a revised version of the manuscript that addresses the points raised during the review process.

We look forward to receiving your revised manuscript.

Kind regards,

Kuo-Cherh Huang

Academic Editor

PLOS ONE

Journal Requirements:

2. Please remove all personal information, ensure that the data shared are in accordance with participant consent, and re-upload a fully anonymized data set.

Additional Editor Comments :

Dear Dr. Zhou,

We appreciate your submission to PLoS ONE. I have received the review reports from two referees with expertise in the field of depressive disorder. Please respond to each comment of the reviewers thoroughly, especially re-visit the Discussion section assiduously as both reviewers had raised fundamental issues in the Discussion. Thank you.

Kuo-Cherh Huang

Academic Editor

Reviewers' comments:

Reviewer's Responses to Questions

**Comments to the Author**

1. Is the manuscript technically sound, and do the data support the conclusions?

Reviewer #1: Yes

Reviewer #2: Yes

2. Has the statistical analysis been performed appropriately and rigorously? 

Reviewer #1: Yes

Reviewer #2: Yes

3. Have the authors made all data underlying the findings in their manuscript fully available?

Reviewer #1: Yes

Reviewer #2: Yes

4. Is the manuscript presented in an intelligible fashion and written in standard English?

Reviewer #1: Yes

Reviewer #2: Yes

5. Review Comments to the Author

Reviewer #1: This is an intelligent review of an important issue.

I only question on sentence in the discussion section:

"Taken together, these results demonstrate no statistically meaningful link between depression and mortality risk, irrespective of whether depression was classified by PHQ-9 scores alone or in combination with antidepressant use."

The study was not designed to study depression and mortality.

The sentence could be phrased more clearly, or deleted entirely.

Reviewer #2: To me it is an lInteresting topic, but I have some comments on the discussion part and others especially on the reference, some parts the reference method is vancouver, and some parts Harvard. As to me it is possible to use one reference style.

1. Does all types antidepressants are safe especially TCAs

2. What are the challenges and Limitations of the study

3. Recommendations of the study

4. Antidepressant Use and All-Cause Mortality in Depressed Individuals, what is ur findings on the mortality in clients with antidepressants?

5. Antidepressant users (n=3,925) had a crude mortality rate of 16.5%, compared to12.2% in non-users (n=2,022). Is proportional ? control i.e non users and antidepressants users. If yes , this results shows that mortality is higher in this users of antidepressants.

6. PLOS authors have the option to publish the peer review history of their article (what does this mean? ). If published, this will include your full peer review and any attached files.

**Do you want your identity to be public for this peer review?** For information about this choice, including consent withdrawal, please see our Privacy Policy .

Reviewer #1: **Yes: ** C T Gualtieri

Reviewer #2: **Yes: ** Gessessew Teklebrhan Gebrehiwot

---

## [Author Response · Author response to Decision Letter 1]

3 Jun 2025

Manuscript ID: PONE-D-25-19212

Title: Antidepressant use and all-cause mortality in depressed individuals: a real-world cohort study’

Dear Dr. Kuo-Cherh Huang and Reviewers,

We sincerely appreciate the time and effort you have dedicated to reviewing our manuscript and providing constructive feedback. We are grateful for the opportunity to revise our work and address the reviewers' comments. In order to facilitate your review, all changes and additions made in the revised manuscript are highlighted in blue. Below, we provide a point-by-point response to each comment, detailing the changes made to the manuscript.

Journal Requirements

1. PLOS ONE Style and File Naming

Response: We have revised our manuscript according to the PLOS ONE style templates, including formatting and file naming requirements. All references have been checked and uniformly formatted in Vancouver style to ensure completeness and accuracy. We have carefully revised Table 1 according to the journal's formatting requirements. To improve space efficiency and maintain clarity, we have consolidated the originally split table into a single presentation. Additionally, we have modified the presentation of binary categorical variables (e.g., Gender) by displaying only one category with its percentage (Male, %), while maintaining all statistical values. This adjustment preserves the complete information while optimizing the table's layout to fit within one page. All data remain unchanged from the original submission. Table S1 (Supplementary Material) has been updated following the same revisions.

2. Data Anonymization

Response: Our study uses the publicly available NHANES database, which does not contain any personally identifiable information. No personal data is included in our submission.

3. Supporting Information Captions

Response: We have now included the captions for all Supporting Information files at the end of the manuscript. All in-text citations to Supporting Information remain accurate and unchanged..

4. Reference List Review

Response: We have carefully reviewed and revised all references to ensure completeness and accuracy, and standardized them to Vancouver style. In response to reviewer concerns regarding tricyclic antidepressants (TCAs), we have added references 26 and 27. The placement of these additions does not alter the sequence of the other references since these two citations come after all the other references.

Reviewers' Comments

Reviewer #1: This is an intelligent review of an important issue.

I only question on sentence in the discussion section:

"Taken together, these results demonstrate no statistically meaningful link between depression and mortality risk, irrespective of whether depression was classified by PHQ-9 scores alone or in combination with antidepressant use."

The study was not designed to study depression and mortality.

The sentence could be phrased more clearly, or deleted entirely.

Response: Thank you very much for your positive comments and for highlighting this important point. We agree with your observation that the sentence in question does not accurately reflect the study design and could be misleading. In accordance with your suggestion, we have deleted the sentence from the discussion section. We believe this change improves the clarity and focus of our manuscript.

Thank you again for your constructive comment, which has helped improve our paper.

Reviewer #2: To me it is an lInteresting topic, but I have some comments on the discussion part and others especially on the reference, some parts the reference method is vancouver, and some parts Harvard. As to me it is possible to use one reference style.

Response: We sincerely appreciate your diligent review and valuable feedback on our manuscript. In response to your concerns regarding reference formatting, we have carefully reviewed all citations and standardized them according to the Vancouver style to ensure consistency. Additionally, we have revalidated and corrected the accuracy of all references.

For example:

Original:

1. World Health Organization. (2023). Depressive disorder (depression). https://www.who.int/news-room/fact-sheets/detail/e-coli [Accessed January 27 2025].

Revised:

1. World Health Organization [Internet]. Depressive disorder (depression). 2023 [cited 2025 Jan 27]. Available from: https://www.who.int/news-room/fact-sheets/detail/depression

Original:

17. Rovin BH, Adler SG, Barratt J, Bridoux F, Burdge KA, Chan TM, et al. KDIGO 2021 Clinical Practice Guideline for the Management of Glomerular Diseases. Kidney International. (2021) 100:S1-S276. doi:10.1016/j.kint.2021.05.021

Revised:

17. Rovin BH, Adler SG, Barratt J, Bridoux F, Burdge KA, Chan TM, et al. KDIGO 2021 clinical practice guideline for the management of glomerular diseases. Kidney Int. 2021;100:S1-S276. doi:10.1016/j.kint.2021.05.021

We have carefully checked and revised all other references line by line to ensure they conform to the Vancouver style. Thank you again for your helpful suggestions.

1. Does all types antidepressants are safe especially TCAs

Response: We sincerely appreciate your thoughtful question regarding the safety of different antidepressant types, particularly TCAs. Your comment highlights an important limitation in our study that deserves further clarification. We understand your concern stems from the fact that different classes of antidepressants have distinct pharmacological profiles and safety considerations. TCAs, in particular, have been associated with greater cardiovascular risks compared to some other antidepressants. Our broad conclusion about antidepressant safety without class-specific analysis could potentially mask important differences in safety profiles.

In our study, we analyzed antidepressants as a collective group due to several methodological constraints. First, many participants were on multiple antidepressants or switched medications during the follow-up period, making class-specific analysis challenging. Second, the NHANES database, while comprehensive, does not provide sufficient granularity on medication dosage, duration, and adherence to conduct robust class-specific analyses. Additionally, stratifying by antidepressant class would have significantly reduced statistical power for each subgroup.

We acknowledge this important limitation and have revised our manuscript to more clearly articulate the scope of our findings. We have modified our conclusion to emphasize that our results reflect the overall safety profile of antidepressants as a collective group, while acknowledging that safety profiles may vary among specific classes.

In the manuscript, we have made the following revisions:

Discussion section:

Add a paragraph (paragraph 3, around line 259-266): "It should be emphasized that our findings reflect the average effect of antidepressant use as a whole and should not be construed as evidence that all classes of antidepressants share identical safety profiles. In this study, antidepressants were analyzed collectively, without differentiation by specific class. However, various classes—such as selective serotonin reuptake inhibitors (SSRIs), serotonin-norepinephrine reuptake inhibitors (SNRIs), tricyclic antidepressants (TCAs), and monoamine oxidase inhibitors (MAOIs)—possess distinct pharmacological characteristics and may present differing safety concerns. For example, TCAs have been associated with a higher risk of cardiovascular adverse events compared to some other antidepressants[26,27]."

Limitations section (around line 310-313):

Original: "Furthermore, the study did not differentiate between different types of antidepressants, limiting our ability to assess the safety of specific antidepressant medications."

Expand: "Fifth, the study did not differentiate between different types of antidepressants, limiting our ability to assess the safety of specific antidepressant classes. This is particularly important as different antidepressants—such as SSRIs, SNRIs, TCAs, and MAOIs—have distinct pharmacological profiles and potentially different safety considerations."

2. What are the challenges and Limitations of the study

Response: We sincerely appreciate your question regarding the challenges and limitations of our study. This is indeed a critical aspect of scientific research that warrants thorough discussion.

We understand that a comprehensive acknowledgment of study limitations is essential for the proper interpretation of research findings. In our manuscript, we have included a limitations section, but we acknowledge that a more detailed discussion would strengthen the paper and provide readers with a clearer understanding of the constraints under which our conclusions should be interpreted.

In our study, we faced several methodological challenges. First, while the NHANES database provides a nationally representative sample with standardized data collection protocols, it has inherent limitations in terms of medication details, such as dosage, duration, and adherence patterns. Second, despite our rigorous statistical approaches to control for confounding, including propensity score matching and inverse probability of treatment weighting, residual confounding from unmeasured variables remains a possibility. Third, the operational definition of depression using PHQ-9 scores or antidepressant use, while pragmatic, may not capture the full spectrum of depressive disorders with the same precision as structured clinical interviews.

We have revised our manuscript to provide a more comprehensive discussion of these limitations and their potential impact on our findings. The expanded limitations section now addresses additional challenges such as the inability to account for changes in depression severity over time, potential selection bias, and the generalizability of our findings to populations outside the United States.

In the manuscript, we have made the following revisions:

Discussion section (around line 266-272):

Add a paragraph acknowledging key methodological challenges: "Additionally, our study faced several methodological challenges worth noting. The assessment of depression using PHQ-9 or antidepressant use, while practical for large epidemiological studies, may not capture the nuanced clinical presentation of depression. Moreover, the NHANES database lacks detailed information on antidepressant dosage, duration, and adherence patterns, which may influence mortality outcomes. These limitations may hinder causal inference, despite the application of robust statistical methods to control for confounders."

Limitations section (around line 299-321):

Original: "Although this study has several strengths, it also has certain limitations. The study's findings are limited to adults aged 20 and older, and may not be applicable to adolescents. Second, although we employed multiple statistical methods to control for confounding factors, there may still be unmeasurable or unknown confounders affecting the results, such as treatment adherence, changes in depression severity, and other undocumented concurrent medications. Furthermore, the study did not differentiate between different types of antidepressants, limiting our ability to assess the safety of specific antidepressant medications. Finally, while NHANES provides a nationally representative sample, extrapolating these results to other countries or regions requires caution, as genetic backgrounds, lifestyle factors, and medical practices may differ across populations."

Revised: "Despite its strengths, our study has several important limitations that warrant consideration. First, our findings are limited to adults aged 20 and older and may not be applicable to adolescents or young adults under 20, a population with unique considerations regarding antidepressant use and safety. Second, although we employed multiple statistical methods to control for confounding factors, residual confounding from unmeasured variables remains possible. Specifically, we lacked detailed information on treatment adherence, changes in depression severity over time, concurrent psychotherapy, and undocumented medications that might interact with antidepressants. Third, the NHANES database, while comprehensive, provides limited information on medication dosage, duration, and patterns of use, which may influence mortality outcomes. Fourth, our operational definition of depression using PHQ-9 scores or antidepressant use, while pragmatic for large-scale epidemiological studies, may not capture the full clinical spectrum of depressive disorders with the same precision as structured clinical interviews. Fifth, the study did not differentiate between different types of antidepressants, limiting our ability to assess the safety of specific antidepressant classes. This is particularly important as different antidepressants—such as SSRIs, SNRIs, TCAs, and MAOIs—have distinct pharmacological profiles and potentially different safety considerations. Sixth, we could not account for potential selection bias in antidepressant prescription, where physicians might preferentially prescribe certain antidepressants to patients with specific health profiles. Finally, while NHANES provides a nationally representative sample of the US population, extrapolating these results to other countries or regions requires caution, as genetic backgrounds, healthcare systems, prescribing practices, and lifestyle factors may differ substantially across populations. These limitations highlight the need for further research with more detailed medication data and longer follow-up periods to understand better the long-term safety of specific antidepressant classes in various subpopulations."

3. Recommendations of the study

Response: Thank you for your valuable suggestion regarding the recommendations of our study. We appreciate the opportunity to elaborate on the practical implications of our findings.

We understand that translating research findings into actionable recommendations is crucial for enhancing the clinical utility and impact of scientific research. While our manuscript briefly discusses some clinical implications, we acknowledge that a more comprehensive and structured presentation of recommendations would strengthen the paper.

Based on our findings that antidepressant use does not significantly impact all-cause mortality in individuals with depression, we have expanded our recommendations for clinical practice and healthcare policy.

We have revised our manuscript to include a dedicated "Recommendations" subsection within the Discussion section, providing more specific and actionable guidance based on our findings. As you rightly noted, while our pooled analysis found no overall mortality risk with antidepressants, certain classes (e.g., TCAs) may carry distinct risks. The original recommendation to prioritize efficacy without mortality concerns might oversimplify clinical decision-making, especially for high-risk patients or specific drug types.

In the manuscript, we have made the following revisions:

Discussion section (around line 276-283):

Original: "For clinical practice, we recommend that physicians focus more on the therapeutic effects of antidepressants rather than being overly concerned about their impact on mortality risk. However, given the heterogeneity of depressed patients, more research is needed for specific subgroups (such as elderly patients and those with multiple comorbidities) to optimize individualized treatment strategies."

Revised: "For clinical practice, while antidepressants as a collective group show no significant mortality risk, clinicians should still consider individual patient factors (e.g., comorbidities, age) and drug-specific safety profiles when prescribing. Pre-treatment safety assessments remain essential, particularly for agents with known risks (e.g., TCAs in cardiovascular disease). In healthcare policy, guidelines for depression management should recognize the overall safety of antidepressants in relation to mortality outcomes, while also emphasizing the significance of appropriate prescribing practices. Mental health literacy programs should address misconceptions about the safety of antidepressants to diminish stigma and overcome barriers

---

## [Decision Letter · Decision Letter 1]

Antidepressant use and all-cause mortality in depressed individuals: a real-world cohort study

PONE-D-25-19212R1

Dear Dr. zhou,

We’re pleased to inform you that your manuscript has been judged scientifically suitable for publication and will be formally accepted for publication once it meets all outstanding technical requirements.

Kind regards,

Kuo-Cherh Huang

Academic Editor

PLOS ONE

Additional Editor Comments (optional):

Reviewers' comments:

Reviewer's Responses to Questions

**Comments to the Author**

1. If the authors have adequately addressed your comments raised in a previous round of review and you feel that this manuscript is now acceptable for publication, you may indicate that here to bypass the “Comments to the Author” section, enter your conflict of interest statement in the “Confidential to Editor” section, and submit your "Accept" recommendation.

Reviewer #1: All comments have been addressed

2. Is the manuscript technically sound, and do the data support the conclusions?

Reviewer #1: Yes

3. Has the statistical analysis been performed appropriately and rigorously? 

Reviewer #1: Yes

4. Have the authors made all data underlying the findings in their manuscript fully available?

Reviewer #1: Yes

5. Is the manuscript presented in an intelligible fashion and written in standard English?

Reviewer #1: Yes

6. Review Comments to the Author

Reviewer #1: The authors have met my concerns. Good to publish. Nice job. I dont know why this APPis not letting me proof and print.

7. PLOS authors have the option to publish the peer review history of their article (what does this mean? ). If published, this will include your full peer review and any attached files.

**Do you want your identity to be public for this peer review?** For information about this choice, including consent withdrawal, please see our Privacy Policy .

Reviewer #1: **Yes: ** C Thomas Gualtieri MD

---

## [Editor Report · Acceptance letter]

PONE-D-25-19212R1

PLOS ONE

Dear Dr. zhou,

I'm pleased to inform you that your manuscript has been deemed suitable for publication in PLOS ONE. Congratulations! Your manuscript is now being handed over to our production team.

Kind regards,

on behalf of

Dr. Kuo-Cherh Huang

Academic Editor

PLOS ONE